# Relationship between Using Fibrate and Open-Angle Glaucoma in Hyperlipidemic Patients: A Population-Based Cohort Study

**DOI:** 10.3390/ijerph19042415

**Published:** 2022-02-19

**Authors:** Yung-En Tsai, Yi-Hao Chen, Chien-An Sun, Chi-Hsiang Chung, Wu-Chien Chien, Ke-Hung Chien

**Affiliations:** 1Department of Ophthalmology, Kaohsiung Armed Forces General Hospital, Kaohsiung 802, Taiwan; zun7511@gmail.com; 2Department of Ophthalmology, Tri-Service General Hospital, National Defense Medical Center, Taipei 114, Taiwan; doc30879@mail.ndmctsgh.edu.tw; 3Department of Public Health, College of Medicine, Fu-Jen Catholic University, New Taipei City 242, Taiwan; 040866@mail.fju.edu.tw; 4Big Data Research Center, College of Medicine, Fu-Jen Catholic University, New Taipei City 242, Taiwan; 5Department of Medical Research, Tri-Service General Hospital, National Defense Medical Center, Taipei 114, Taiwan; g694810042@gmail.com; 6School of Public Health, National Defense Medical Center, Taipei 114, Taiwan; 7Taiwanese Injury Prevention and Safety Promotion Association, Taipei 114, Taiwan; 8Graduate Institute of Life Sciences, National Defense Medical Center, Taipei 114, Taiwan

**Keywords:** fibrate, open-angle glaucoma, cohort study

## Abstract

Background: This study aimed to investigate the associated risk between using fibrate and open-angle glaucoma (OAG) in hyperlipidemic patients from the National Health Insurance Research Database (NHIRD). Methods: We collected data over a 16-year period from the NHIRD, and used the Fisher’s exact test and Pearson chi-square test to analyze variables. Adjusted hazard ratios (aHR) were used to examine the risk factors for disease development. We applied Kaplan–Meier analysis to compare the cumulative incidence of OAG. Results: A total of 10,011 patients using fibrate were enrolled in the study cohort, and 40,044 patients not using fibrate were enrolled in the control cohort. The incidence of OAG was lower in the study cohort than in the control cohort (aHR = 0.624, *p* = 0.007). The overall incidence of OAG was 463.02 per 100,000 person-years in the study cohort and 573.65 per 100,000 person-years in the control cohort. We used the Kaplan–Meier method to calculate the cumulative risk of developing OAG. The results revealed that after using fibrate for over seven years, the study cohort had a greatly lower rate of developing OAG than the control cohort (log-rank test *p* = 0.050). Conclusions: Our studies showed that using fibrate for over seven years may lead to a lower risk of OAG in patients with hyperlipidemia. Nevertheless, further prospective studies that comprehensively investigate the relationship between using fibrate and OAG are needed.

## 1. Introduction

Glaucoma is a group of diseases characterized by optic neuropathy that is consistent with remodeling of the connective tissue elements of the optic nerve head, as well as the loss of neural tissue associated with the development of distinctive patterns of visual dysfunction. The prevalence and disease burden of glaucoma has recently increased in Asia, including Taiwan [1]. Hyperlipidemia is significantly related with an increased risk of glaucoma, and hyperlipidemia and increased blood lipid levels are related with increased intraocular pressure (IOP) [2]. Fibrates are peroxisome proliferator-activated receptor (PPAR)-α agonists used for the treatment of dyslipidemia [3]. PPARα has been broadly studied in the retina, and has been proven to be related with the neuro-protection, angiogenesis, and regulation of circadian rhythms [4]. In experimental models of cerebral ischemia and neurodegenerative diseases, it has been shown that PPARα activation has a neuroprotective effect [5]. However, it has never evaluated the neuroprotective effect of fibrates within the context of open-angle glaucoma (OAG).

This study aimed to investigate the associated risk between using fibrates and the prevalence of OAG in our population using data collected from the National Health Insurance Research Database (NHIRD) of Taiwan. As far as we know, no study has investigated this issue in the Taiwanese population.

## 2. Materials and Methods

### 2.1. Research Database

The National Health Insurance (NHI) program was launched in Taiwan in 1995 and is a single-payer, nationwide health insurance program. The NHI contains almost 99% of the Taiwanese population, and currently contains the health expenses of probably 23 million people. The claims data related to patients from the NHI program is contained in the NHIRD. The basic information of these data, such as age, sex, diagnosis, insurance premium, location, season, level of care, and urbanization level, can be collected from the NHIRD. Additionally, an electronic format of these data are available for conducting statistical studies. Thus, we analyzed this database to investigate the relationship between fibrates and OAG.

### 2.2. Study Participants

This study was a retrospective cohort study. As demonstrated in Figure 1, we identified patients by NHIRD with clinical diagnoses of OAG and prescribed fibrates based on the International Classification of Disease, 9th Revision, Clinical Modification (ICD-9-CM) code (ICD-9-CM code for glaucoma, 365; ATC code for fenofibrate, C10AB05; gemfibrozil, C10AB04). The inclusion criterion of this study is fibrate use and the exclusion criteria are fibrate before index date, glaucoma before tracking, statin/Pravafen in the study period, laser or surgery treatment of glaucoma before index date, without tracking, age less than 18 years and gender unknown. We also identified, differentiated, and analyzed the comorbidities of these patients with OAG by ICD-9-CM codes. Since the removed diseases were also variables in this study, we substituted the Charlson comorbidity index (CCI) for the revised Charlson Comorbidity Index (CCI_R) (CCI excluding hypertension [HTN], diabetes mellitus [DM], coronary artery disease [CAD], pneumonia, liver disease, and renal disease). From the year 2000 to 2015, this study recognized 48,956 individuals who fit the inclusion criteria. However, 38,945 patients were expelled based on the exclusion criteria. Eventually, the study cohort included 10,011 individuals. The control cohort was collected using the same criteria as the study cohort using four-fold propensity score matching by age, sex, and index date, resulting in 40,044 individuals being included in the comparison cohort. During the 16-year period, 2235 patients were confirmed with OAG, involving 376 individuals in the study cohort and 1859 individuals in the control cohort (Figure 1). This study protocol corresponded to the ethical guidelines of the 1975 Declaration of Helsinki.

### 2.3. Ethical Considerations

The NHIRD encodes personal patient information to protect privacy. To access NHIRD data, patient consent was not necessary. This study was approved by the Institutional Review Board of the Tri-Service General Hospital and the requirement for consent was waived (TSGHIRB No.: B-110-37).

### 2.4. Statistical Analysis

The Fisher’s exact test and Pearson chi-square test were used to calculate the differences in variables, such as age group, sex, and insurance premium, with statistical significance defined at *p* < 0.05. After adjusting the variables, we used univariate and multivariate Cox regression analyses to evaluate the adjusted hazard ratio (aHR) for the effect of using fibrate on preventing OAG. Then, we performed Kaplan–Meier analysis to predict the cumulative incidence of OAG in these two cohorts. We used SPSS 22nd edition to perform all statistical analyses (SPSS Inc., Chicago, IL, USA).

## 3. Results

### 3.1. Baseline Characteristics of the Subjects

Table 1 reveals the demographic characteristics of the cohorts. There were no significant differences in sex between the two groups. The mean age was 49.98 ± 18.25 years in the study cohort and 49.71 ± 18.03 years in the control cohort. The difference was not statistically significant (*p* = 0.181). In the comorbidity comparison, patients using fibrates had higher rates of DM, HTN, CAD, heart failure (HF), cardiovascular disease (CVD), renal disease, injury, tumor, and obesity (Figure 2). The CCI_R value was 1.06 ± 1.24 in the study cohort and 0.99 ± 1.18 in the control cohort (*p* < 0.001). Additionally, more individuals in the study cohort lived in eastern Taiwan, offshore islands, and the lowest urbanized areas, with most of them receiving therapy in physician clinics than in the control cohort (*p* < 0.001).

### 3.2. Fibrate Lowers the Risk of Developing OAG

We calculated the cumulative risk of developing OAG using the Kaplan–Meier method (Figure 3). The results revealed that after using fibrate for over seven years, the study cohort had a greatly lower rate of developing OAG than the control cohort (log-rank test *p* = 0.05). At the endpoint in the study cohort, the cumulative risk of OAG declined every year to 3.75% (376/10,011 individuals), and to 4.64% (1859/40,044 individuals) in the control cohort. In addition, the difference between two groups was obvious since the seventh year of follow-up (*p* < 0.05 since the seventh year).

### 3.3. The Risk Factors for OAG

Table 2 reveals the risk factors for OAG in the Cox regression analysis. The crude hazard ratio (HR) for OAG was 0.602 (95% CI = 0.387–0.955) times lower in the using fibrate group. After adjustment for covariates, the significance remained (adjusted HR = 0.624, 95% CI = 0.401–0.960). In both crude and adjusted HR analyses, patients with DM, HTN, CVD, CAD, obesity, and renal disease had a remarkably higher risk of developing OAG. The characteristics of the patients with other comorbidities, such as sex and other chronic diseases, were not obviously related with OAG diagnosis based on the obtained hazard ratios (all *p* > 0.05). Figure 4 revealed Forest plots of crude and adjusted HR for factors for OAG evaluated in Table 2.

### 3.4. The Total Incidence of OAG with and without Using Fibrate in the Subgroup Analysis

Comparing patients with and without using fibrate in the subgroup analysis (Table 3), the total incidence of OAG was 463.02 per 100,000 person-years in the study cohort and 573.65 per 100,000 person-years in the control cohort. Both male and female patients with using fibrate had a decreased risk of developing OAG (aHR = 0.630 in men and 0.618 in women, *p* = 0.014 and 0.001, respectively).

## 4. Discussion

This study included 10,011 patients in the study cohort and 40,044 patients in the control cohort. It shows that the rate of developing OAG was greatly lower in the study cohort than in the control cohort. Kaplan–Meier analysis also demonstrated that the cumulative risk of developing OAG was greatly decreased in the study cohort since the seventh year of using the fibrate. As far as we know, the relationship between using fibrates and OAG in humans has not been evaluated in previous studies.

In the study by Gupta et al. [6], the administration of clofibrate, a lipid-lowering agent, prevented a marked rise in the IOP before producing OAG in rabbits and, when given after producing OAG, caused a decline in IOP, thereby indicating that the drug could be of prophylactic and therapeutic value for OAG. However, contradictory results have been noted in a study involving human subjects [7]. In the study by James et al. [7], using fibrate did not lead to significant alterations in intraocular pressure or coefficients of outflow, and fibrates did not appear to be a useful addition to the drugs being used for OAG therapy. In our study, we first reported that using fibrate resulted in a 0.624-fold decreased risk of OAG (adjusted hazard ratio [HR] = 0.624, 95% CI = 0.401–0.960) compared to not using fibrate. Our results conflict with those in the previous literature. Nevertheless, our findings cannot be compared with other studies due to the different study designs and methodologies.

In the present study, patients with DM (aHR = 2.401, *p* ≤ 0.001), HTN (aHR = 1.762, *p* ≤ 0.001), CAD (aHR = 1.420, *p* ≤ 0.001), CVD (aHR = 1.594, *p* ≤ 0.001), renal disease (aHR = 1.562, *p* = 0.002), and obesity (aHR = 2.000, *p* ≤ 0.034) were at higher risk of OAG development. DM, HTN, CAD, CVD, renal disease, and obesity have been disclosed to be risk factors for OAG in previous studies [8,9,10,11,12,13]. Y. Jung et al. [8] conducted a population-based cohort study, and showed that DM was a great risk factor for OAG. Kuang et al. [9] conducted a population-based case-control study, and showed that OAG was associated with pre-existing HTN. Chen et al. [10] conducted a population-based retrospective cohort study, and showed that people with OAG may experience ischemic heart disease more often than those without glaucoma. De Escalona-Rojas et al. [11] conducted a cross-sectional study, and showed that there was a statistically great relation between the presence of CVD and/or cardiovascular risk factors with glaucomatous disease. Park et al. [12] conducted a population-based longitudinal cohort study and showed that OAG increased the risk of subsequent chronic kidney disease in the general population. Jung et al. [13] conducted a population-based, longitudinal prospective cohort study, and showed that obesity and metabolic health status were greatly related with an increased risk of OAG incidence. Consistent with the findings of previous studies [8,9,10,11,12,13], our study found that DM, HTN, CAD, CVD, renal disease, and obesity were risk factors for OAG development (Table 2).

Furthermore, considering using fibrate as a presumed indicator, we performed a subgroup analysis in our study (Table 3). Patients using fibrate had a decreased risk of developing OAG than those in the control cohort, in spite of sex group analysis, with this difference being statistically significant (*p* < 0.050). It is possible that the study cohort included patients who had maladaptive endocrine conditions. Their unstable systemic conditions may have led to hyperlipidemia and induced multiple organ dysfunction, manifesting as DM, HTN, CAD, CVD, renal disease, obesity, and OAG.

Hyperlipidemia is greatly related with an increased risk of OAG. In fact, increased blood lipid levels are associated with increased intraocular pressure [2]. Fibrates are an extensively used class of lipid-lowering agents, which activate peroxisome proliferator-activated receptors (PPARs) [14]. PPARα is the molecular target of fibrates. Activation of PPARα remarkably declines production of hepatic triglyceride and improves clearance of plasma triglyceride, resulting in a clinically great reduction in plasma triglyceride levels. Additionally, plasma high-density lipoprotein cholesterol levels are increased upon PPARα activation [15]. PPARα activation has anti-fibrotic, anti-angiogenic, and anti-inflammatory effects, and can also modulate oxidative stress responses in different organs (including the eyes). The pathologic mechanisms of OAG involve neo-angiogenesis, inflammation, and oxidative stress-mediated cell death, but evidence is accumulating regarding the potential benefits of PPAR modulation in preventing or ameliorating eye pathologies [16]. According to the aforementioned studies, fibrates are an alternative therapy given for OAG patients, with their results finding that OAG may be associated with PPARα activity. In our study, using fibrate over seven years has been associated with lower OAG risk in patients with hyperlipidemia.

Our study has some limitations. First of all, our study was a retrospective cohort study. Next, database research studies lack data regarding imaging examination findings, such as from fundoscopy, perimetry, and optical coherence tomography findings, that would have helped confirm the patients’ diagnosis of OAG. Third, our population in this study was Han Chinese in Taiwan. Therefore, other ethnic groups may not be applicable to our study results. Lastly, there is selection bias since the cohort enrollment was limited to patients with OAG and the control cohort in our study. Nonetheless, there are some strengths in our study. First, since Taiwan started the NHI system in 1995, longitudinal data analysis was allowed to compare the cumulative incidence of OAG between the study cohort and control cohort over a long-term study period. In addition, all citizens in Taiwan are obligated to enroll in the NHI, which made the coverage rate almost 99% [17]. Therefore, the data we collected in our study were from a population-based, nationwide database in Taiwan. However, we are not sure about the exact mechanism which may lead to a lower OAG risk after using fibrates for over seven years. Further detailed studies should be conducted to clarify this correlation.

## 5. Conclusions

In conclusion, the results indicating that using fibrates for over seven years can lower OAG risk is proved by the statistical information from the NHIRD. This finding can be validated and confirmed by determining the molecular mechanisms and the pathogenic pathways of OAG affected by fibrates. If the protective effect of fibrates is confirmed in future studies, they may represent a novel therapeutic strategy for treating OAG.

## Figures and Tables

**Figure 1 ijerph-19-02415-f001:**
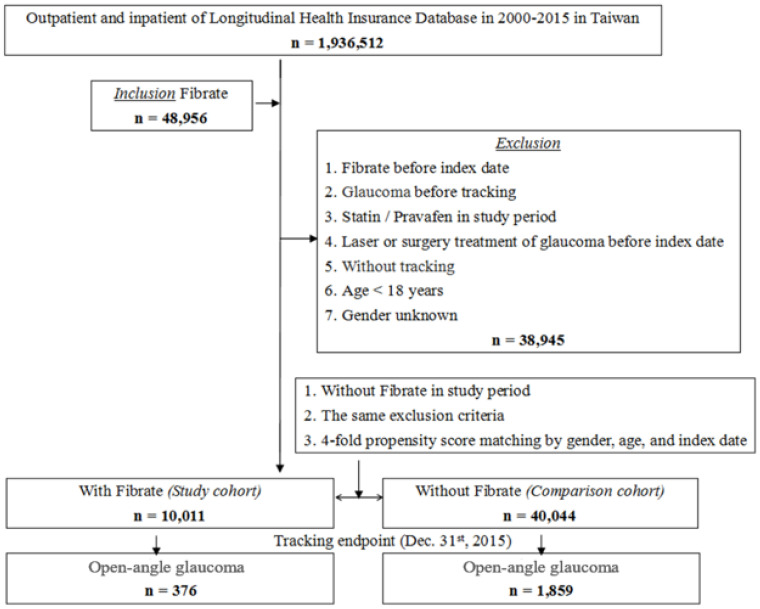
Flowchart of the study sample selection from the NHIRD in Taiwan.

**Figure 2 ijerph-19-02415-f002:**
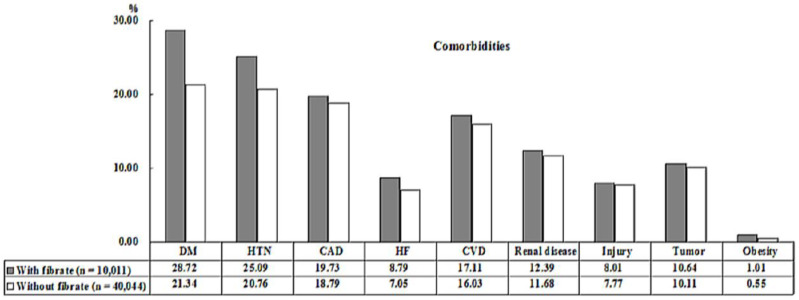
Bar chart of the comorbidities of the study population from Table 1.

**Figure 3 ijerph-19-02415-f003:**
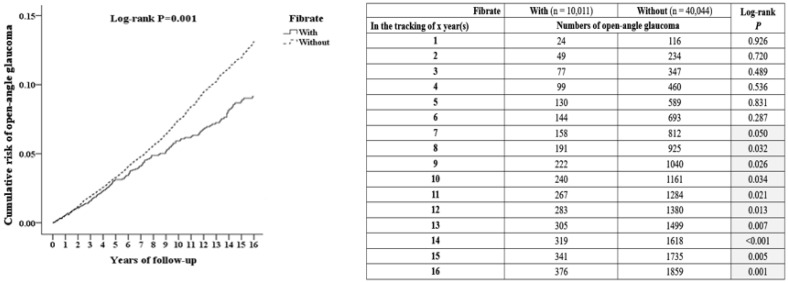
Kaplan-Meier analysis of the cumulative risk of open-angle glaucoma among patients aged 18 and over, stratified by using fibrate with the log-rank test.

**Figure 4 ijerph-19-02415-f004:**
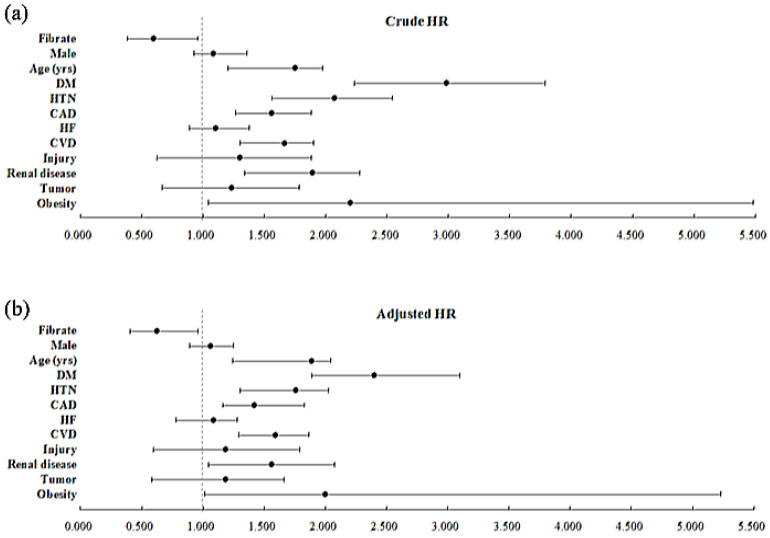
Forest plots of crude (**a**) and adjusted (**b**) hazard ratio for factors for open-angle glaucoma evaluated in Table 2.

**Table 1 ijerph-19-02415-t001:** Baseline characteristics of the study population.

Fibrate	Total		With		Without		*p*
Variables	n	%	n	%	n	%	
Total	50,055		10,011	20.00	40,044	80.00	
Gender							0.999
Male	27,855	55.65	5571	55.65	22,284	55.65	
Female	22,200	44.35	4440	44.35	17,760	44.35	
Age (years)	49.76 ± 18.07		49.98 ± 18.25		49.71 ± 18.03		0.181
DM							<0.001
Without	38,636	77.19	7136	71.28	31,500	78.66	
With	11,419	22.81	2875	28.72	8544	21.34	
HTN							<0.001
Without	39,231	78.38	7499	74.91	31,732	79.24	
With	10,824	21.62	2512	25.09	8312	20.76	
CAD							0.032
Without	40,557	81.02	8036	80.27	32,521	81.21	
With	9498	18.98	1975	19.73	7523	18.79	
HF							<0.001
Without	46,350	92.60	9131	91.21	37,219	92.95	
With	3705	7.40	880	8.79	2825	7.05	
CVD							0.009
Without	41,922	83.75	8298	82.89	33,624	83.97	
With	8133	16.25	1713	17.11	6420	16.03	
Renal disease							0.049
Without	44,138	88.18	8771	87.61	35,367	88.32	
With	5917	11.82	1240	12.39	4677	11.68	
Injury							0.424
Without	46,141	92.18	9209	91.99	36,932	92.23	
With	3914	7.82	802	8.01	3112	7.77	
Tumor							0.119
Without	44,941	89.78	8946	89.36	35,995	89.89	
With	5114	10.22	1065	10.64	4049	10.11	
Obesity							<0.001
Without	49,732	99.35	9910	98.99	39,822	99.45	
With	323	0.65	101	1.01	222	0.55	
CCI_R	1.00 ± 1.19		1.06 ± 1.24		0.99 ± 1.18		<0.001
Location							<0.001
Northern Taiwan	15,118	30.20	3012	30.09	12,106	30.23	
Middle Taiwan	13,522	27.01	2511	25.08	11,011	27.50	
Southern Taiwan	14,683	29.33	2804	28.01	11,879	29.66	
Eastern Taiwan	5256	10.50	1375	13.73	3881	9.69	
Outlets islands	1476	2.95	309	3.09	1167	2.91	
Urbanization level							0.083
1 (The highest)	14,644	29.26	2886	28.83	11,758	29.36	
2	15,284	30.53	3015	30.12	12,269	30.64	
3	9974	19.93	1988	19.86	7986	19.94	
4 (The lowest)	10,153	20.28	2122	21.20	8031	20.06	
Level of care							<0.001
Medical center	10,389	20.76	2024	20.22	8365	20.89	
Regional hospital	11,874	23.72	1988	19.86	9886	24.69	
Local hospital	12,622	25.22	2498	24.95	10,124	25.28	
Physician Clinics	15,170	30.31	3501	34.97	11,669	29.14	

*p*, Chi-square/Fisher exact test on category variables and *t* test on continue variables. CCI_R, Charlson comorbidity index removed HF, DM, HTN, CVD, and CAD; CAD, coronary artery disease; CVD, cardiovascular disease; DM, diabetes mellitus; HF, heart failure; HTN, hypertension.

**Table 2 ijerph-19-02415-t002:** Factors for open-angle glaucoma evaluated with Cox regression.

Variables	Crude HR	95% CI	95% CI	*p*	Adjusted HR	95% CI	95% CI	*p*
Fibrate								
Without	Reference				Reference			
With	0.602	0.387	0.955	0.001	0.624	0.401	0.960	0.007
Gender								
Male	1.088	0.927	1.356	0.186	1.065	0.894	1.246	0.203
Female	Reference				Reference			
Age (yrs)	1.752	1.201	1.976	<0.001	1.892	1.244	2.045	<0.001
DM								
Without	Reference				Reference			
With	2.989	2.235	3.786	<0.001	2.401	1.888	3.097	<0.001
HTN								
Without	Reference				Reference			
With	2.075	1.562	2.544	<0.001	1.762	1.301	2.026	<0.001
CAD								
Without	Reference				Reference			
With	1.562	1.268	1.885	<0.001	1.420	1.165	1.831	<0.001
HF								
Without	Reference				Reference			
With	1.103	0.892	1.378	0.188	1.086	0.781	1.279	0.276
CVD								
Without	Reference				Reference			
With	1.664	1.304	1.904	<0.001	1.594	1.289	1.862	<0.001
Injury								
Without	Reference				Reference			
With	1.303	0.624	1.880	0.486	1.186	0.597	1.789	0.571
Renal disease								
Without	Reference				Reference			
With	1.894	1.341	2.275	<0.001	1.562	1.048	2.076	0.002
Tumor								
Without	Reference				Reference			
With	1.234	0.670	1.786	0.611	1.189	0.583	1.660	0.679
Obesity								
Without	Reference				Reference			
With	2.201	1.046	5.484	0.005	2.000	1.017	5.231	0.034

HR, hazard ratio; CI, confidence interval; Adjusted HR, adjusted variables listed in the table.

**Table 3 ijerph-19-02415-t003:** Factors for open-angle glaucoma stratified by variables listed in the table evaluated with Cox regression.

Fibrate	With			Without (Reference)			With vs. Without (Reference)			
Stratified	Events	PYs	Rate (per 10^5^ PYs)	Events	PYs	Rate (per 10^5^ PYs)	Adjusted HR	95% CI	95% CI	*p*
Total	376	81,205.68	463.02	1859	324,065.72	573.65	0.624	0.401	0.960	0.007
Gender										
Male	190	45,189.96	420.45	931	180,339.64	516.25	0.630	0.406	0.969	0.014
Female	186	36,015.73	516.44	928	143,726.08	645.67	0.618	0.398	0.951	0.001

PYs, person-years; Adjusted HR, adjusted hazard ratio (adjusted for the variables listed in Table 3); CI, confidence interval.

## Data Availability

Data available in a publicly accessible repository. The data presented in this study are openly available in FigShare at https://figshare.com/s/15bd22eea7e8e207d3b5, https://figshare.com/s/9bb3518d46c18dbf2014, https://figshare.com/s/84b432a389fb7b5ac39c (accessed on 17 February 2022).

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
