# Peer review of "Relationship between Using Fibrate and Open-Angle Glaucoma in Hyperlipidemic Patients: A Population-Based Cohort Study"

_ijerph, 2022, doi:10.3390/ijerph19042415_

Round 1

Reviewer 1 Report

In this article Tsai et al have shown a novel and very interesting correlation between fibrate and open-angle glaucoma in lipidemic patients. This study will incite researchers to further investigate and validate this interesting association. Authors have carefully designed the study and the manuscript is well-written.

1) Author might want to point out the relation between glaucoma and OAG importance in the introduction part with citations. If there is any literature that suggest relationship between hyperlipidemia and glaucoma to set up the background a little more clear to the reader.

2) Flow chart in the method section is very helpful and easy to follow. But the inclusion and exclusion criteria is missing in the text itself. Is there any data available about the IOP for the OAG subjects between the study and control group. This might be important to look at and compare.

3) Can the authors put subsections for results like: 3.1) Baseline characteristics of the subjects, 3.2) Fibrate lowers the risk of developing OAG and so on.

4) Graphical representation for table 1 showing the comparison for DM, HTN, CAD, HF, CVD, renal disease, injury, tumor and obesity might be helpful to the reader.

5) Fig.2- Kaplan-Meier plot clearly shows a strong evidence that use of fibrate reduces the OAG development. However, a graphical representation of crude and adjusted HR for DM, HTN, CAD, HF, CVD, renal disease, injury, tumor and obesity in table 2 can be helpful to interpret the data easily.

6) Authors might need to take care of the use of different fonts throughout like line192-195 etc.

Author Response

[2022.02.12]

Thank you for allowing us to submit a revised draft of our manuscript entitled,” Relationship between using fibrate and open-angle glaucoma in hyperlipidemic patients: A population-based cohort study.” to International Journal of Environmental Research and Public Health. The previous manuscript ID is: ijerph-1538365. We also appreciate the time and effort you and each of the Reviewers have dedicated to providing constructive feedback.

Our responses to the Reviewers’ comments and suggestions are described below in a point-to-point manner. Appropriate changes, suggested by the Reviewers, has been introduced to the manuscript.

Point 1: Author might want to point out the relation between glaucoma and hyperlipidemia importance in the introduction part with citations. If there is any literature that suggest relationship between hyperlipidemia and glaucoma to set up the background a little more clear to the reader.

Response 1: We appreciate your suggestion. We have cited a meta-analysis which pointed out the relation between glaucoma and hyperlipidemia and revised paragraph in the introduction section.

Reference

  1. Shiming W, Xianyi B. Hyperlipidemia, blood lipid level, and the risk of glaucoma: a meta-analysis. Invest Ophthalmol Vis Sci. 2019 Mar 1;60(4):1028-1043.

Point 2: Flow chart in the method section is very helpful and easy to follow. But the inclusion and exclusion criteria is missing in the text itself. Is there any data available about the IOP for the OAG subjects between the study and control group. This might be important to look at and compare.

Response 2: We appreciate your suggestion. We have added inclusion and exclusion criteria in the “2.2. Study participants” part.

In National Health Insurance Research Database of Taiwan, we can track that patient had done the examination of IOP, but the data of IOP was unavailable in this database. Therefore, we can’t provide the IOP data of the OAG subjects between the study and control group.  That is the limitation of our study.

Point 3: Can the authors put subsections for results like: 3.1) Baseline characteristics of the subjects, 3.2) Fibrate lowers the risk of developing OAG and so on.

Response 3: We appreciate your suggestion. We have added subsections for results.

Point 4: Graphical representation for table 1 showing the comparison for DM, HTN, CAD, HF, CVD, renal disease, injury, tumor and obesity might be helpful to the reader.

Response 4: We appreciate your suggestion. We have added Figure 2 for the comparison of the comorbidities of the study population from Table 1, which presented by %.

Point 5: Fig.2- Kaplan-Meier plot clearly shows a strong evidence that use of fibrate reduces the OAG development. However, a graphical representation of crude and adjusted HR for DM, HTN, CAD, HF, CVD, renal disease, injury, tumor and obesity in table 2 can be helpful to interpret the data easily.

Response 5: We appreciate your suggestion. We have added Figure 4 (Forest plots) of crude and adjusted hazard ratio for factors for open-angle glaucoma evaluated in table 2.

Point 6: Authors might need to take care of the use of different fonts throughout like line192-195 etc.

Response 6: We appreciate your suggestion. We have checked the whole manuscript and adjusted to the same font.

Thanks a lot again and your comments and suggestions really helped us to improve our manuscript better. We hope that our manuscript will be acceptable for publication in International Journal of Environmental Research and Public Health.

We look forward to hearing from you.

Sincerely,

Ke-Hung Chien, MD, PhD

Department of Ophthalmology, Tri-Service General Hospital, National Defense Medical Center, Taipei, Taiwan.

No. 325, Chenggong Rd., Sec. 2, Neihu, Taipei 114, Taiwan.

Tel: +886-2-87923311

Fax: +886-2-87927104

E-mail: yred8530@gmail.com

Reviewer 2 Report

The manuscript is well organized according to rules. The aim is clear. Methods and results are well presented. Results are very interesting and provides the occasion for future studies. 

Glaucoma is one of the leading causes of blindness worldwide. For that reason, studies aimed to investigate different ways to decrease incidence and prevent progression of the disease are very interesting and useful.

The main strengths of the manuscript are:

- this is one of the first studies that evaluate associated risk between using fibrate and prevalence of OAG in very large population /10,011 patients in the study cohort and 40,044 patients in the control cohort/   for long period of time – 7 years

- study is well organized, using appropriate statistical methods

- results are well presented

- conclusion that “both male and female patients with using fibrate had a decreased risk of developing OAG”  is very interesting, with practical meaning

- authors draw attention to the limitations of their study

My advise to authors is to continue their work with prospective study aimed to assess influence of fibrates on open-angle glaucoma progression. 

Author Response

[2022.02.12]

Thank you for your approval of our manuscript entitled,” Relationship between using fibrate and open-angle glaucoma in hyperlipidemic patients: A population-based cohort study.” to International Journal of Environmental Research and Public Health. The previous manuscript ID is: ijerph-1538365. We also appreciate the time and effort you and each of the Reviewers have dedicated to providing constructive feedback.

Thanks a lot again and your comments and suggestions really encouraged us to improve our manuscript better. We hope that our manuscript will be acceptable for publication in International Journal of Environmental Research and Public Health.

We look forward to hearing from you.

Sincerely,

Ke-Hung Chien, MD, PhD

Department of Ophthalmology, Tri-Service General Hospital, National Defense Medical Center, Taipei, Taiwan.

No. 325, Chenggong Rd., Sec. 2, Neihu, Taipei 114, Taiwan.

Tel: +886-2-87923311

Fax: +886-2-87927104

E-mail: yred8530@gmail.com

Reviewer 3 Report

This is an interesting manuscript on a well conducted study of relationship between fibrate use and open angle glaucoma. The major strengths of presented study are the wide study period (16 years) and number of participants (10011). I believe this manuscript adds to the body of literature on the matter of glaucoma, especially in hyperlipidemic patients. Moreover, this methodology is appropriate and well described, enables other researchers to use. The figures and tables are well designed. The references are up to date. Therefore, I believe this study is appropriate for publication.

The main question addressed by this research is the relationship between fibrate use and open angle glaucoma.

I believe this topic is relevant in the field. This study adds to the subject area as study of this relationship is sparse, and presented study has wide study period and large sample. I do not have any suggestions for authors in order for them to improve their methodology or presented manuscript.

Moreover, the conclusions are consistent with the evidence and arguments
presented and they address the main question posed. References are appropriate,

Lastely, tables and figure are appropriate and do not require any improvements, 

Author Response

(The authors gave the same response as above.)

Round 2

Reviewer 1 Report

Authors have addressed the concerns raised earlier. The manuscript presents a very interesting correlation between fibrate use and low risk of OAG in hyperlipidemia.